# Surfing the Waves: Differences in Hospitalised COVID-19 Patients across 4 Variant Waves in a Belgian University Hospital

**DOI:** 10.3390/v15030618

**Published:** 2023-02-23

**Authors:** Lucie Seyler, Els Van Nedervelde, Diederik De Cock, Claudia Mann, Karen Pien, Sabine D. Allard, Thomas Demuyser

**Affiliations:** 1Infectiology Research Group, Infectious Diseases Department, Universitair Ziekenhuis Brussel, Vrije Universiteit Brussel (VUB), 1090 Brussels, Belgium; 2Biostatistics and Medical Informatics Research Group, Department of Public Health, Faculty of Medicine and Pharmacy, Vrije Universiteit Brussel (VUB), 1090 Brussels, Belgium; 3Medical Registration, Universitair Ziekenhuis Brussel, 1090 Brussels, Belgium; 4Department of Microbiology and Infection Control, Universitair Ziekenhuis Brussel, Vrije Universiteit Brussel (VUB), 1090 Brussels, Belgium; 5AIMS Lab, Center for Neurosciences, Faculty of Medicine and Pharmacy, Vrije Universiteit Brussel (VUB), 1090 Brussels, Belgium

**Keywords:** COVID-19, SARS-CoV-2, pandemic, inpatients, disease severity, registry data

## Abstract

The unprecedented COVID-19 pandemic took the form of successive variant waves, spreading across the globe. We wanted to investigate any shift in hospitalised patients’ profiles throughout the pandemic. For this study, we used a registry that collected data automatically from electronic patient health records. We compared clinical data and severity scores, using the National Institute of Health (NIH) severity scores, from all patients admitted for COVID-19 during four SARS-CoV-2 variant waves. Our study concluded that patients hospitalised for COVID-19 showed very different profiles across the four variant waves in Belgium. Patients were younger during the Alpha and Delta waves and frailer during the Omicron period. ‘Critical’ patients according to the NIH criteria formed the largest fraction among the Alpha wave patients (47.7%), while ‘severe’ patients formed the largest fraction among Omicron patients (61.6%). We discussed host factors, vaccination status, and other confounders to put this into perspective. High-quality real-life data remain crucial to inform stakeholders and policymakers that shifts in patients’ clinical profiles have an impact on clinical practice.

## 1. Introduction

The end of 2019 was accompanied by the emergence of the unprecedented pandemic of Severe Acute Respiratory Syndrome-Coronavirus-2 (SARS-CoV-2). The resulting disease, COVID-19, was associated with a significant increase in morbidity and mortality [1], with repercussions on many healthcare systems. Pneumonia was the predominant clinical presentation of COVID-19 [2], but other major complications were described early on, such as thrombotic events and inflammatory storms in severe cases [3,4].

It soon appeared that the SARS-CoV-2 virus was mutating fast: the initial virus, reported from Wuhan, China, was replaced by successive variants of concern (VOC), causing epidemic waves across the globe. The VOCs were given letters of the Greek alphabet: Alpha, Beta, Gamma, Delta, and Omicron [5]. The scientific community is still collecting essential real-world observational data to understand the pandemic in detail and be better prepared for future ones.

Many published studies have focused on data spanning one or a couple of COVID-19 waves at a time [6,7,8], but data on the evolution of COVID-19 across waves are scarce [9]. Our study was set up to analyse the clinical characteristics and severity of patients hospitalised for COVID-19 during the COVID-19 pandemic in a Belgian university hospital. We wanted to investigate potential shifts in patients’ profiles across the four most significant epidemic waves in the country.

## 2. Materials and Methods

### 2.1. Data Sources and Ethical Clearance

Patients were included in the UZB SARI-Registry. This is an exhaustive and automated registry capturing clinical data on patients admitted to the Brussels University Hospital (UZ Brussel) with COVID-19 and/or symptoms of severe acute respiratory infection (‘SARI’). It is hosted in REDCap (Research Electronic Data Capture), a secure, web-based software platform designed to support data capture for research purposes [10]. A few additional data were obtained by bulk extraction of parameters from patients’ electronic files by our information technology (IT) system or by manual retrieval. The data spanned each patient’s entire admission.

The study was approved by the UZ Brussel ethics committee (B.U.N. 1432021000456) on March 30, 2022, and was conducted in line with the Helsinki requirements. The UZB SARI-Registry structure has been assigned the following intellectual property number: ‘i-DEPOT 137464’ by the BBIE bureau.

### 2.2. Study Design, Setting, and Duration

Our study is a monocentric observational study using prospectively collected data from the UZB SARI-Registry. The UZ Brussels has a catchment area spanning Brussels and parts of the surrounding Flanders and Wallonia regions. The study period spans the whole COVID-19 pandemic in Belgium, from March 2020 up to February 2022.

### 2.3. Inclusion Periods

The inclusion periods were defined as the periods in which one particular strain of the SARS-CoV-2 virus accounted for 80% or more of the circulating viruses in Belgium [11]. We identified four periods of interest, shown below, based on the Belgian epidemiology; the Beta VOC did not reach 80% dominance at any point and was therefore not included.

Period 1 (50 weeks long): pre-Variant Of Concern (‘pre-VOC’): from 03/02/2020 until the end of week 2 (17/01/2021);Period 2 (9 weeks long): ‘Alpha’ variant, from the start of week 11 (15/03/2021) until the end of week 20 (23/05/2021);Period 3 (23 weeks long): ‘Delta’ variant, from the start of week 27 (05/07/2021) until the end of week 50 (19/12/2021);Period 4 (6 weeks long): ‘Omicron’ variant (BA1), from the start of week 1 (03/01/2022) until the end of week 6 (13/02/2022).

### 2.4. Patient Selection

All adult patients (≥18 years) hospitalised for PCR-confirmed COVID-19 as the main reason for admission during one of the periods of interest described above and for >24 h were included in the study. To ensure that COVID-19 was the main reason for admission, we cross-checked all patients’ ICD-10 diagnostic codes for the ‘main diagnosis’. The patient selection steps are summarised in Figure 1.

### 2.5. Measurements

#### 2.5.1. Vaccination Status

The variable ‘vaccination status’ refers to the number of doses of the COVID-19 vaccine a patient received. ‘Partially vaccinated’ refers to patients who have received only one dose of a 2-dose vaccination scheme (Pfizer^®^, Moderna^®^ or Astra Zeneca^®^ vaccines in Belgium); fully-vaccinated means having received the Janssens vaccine or two doses of a 2-dose vaccine at least 2 weeks before the start of symptoms. Having had a booster means having received one booster after a full vaccination. The vaccination rollout started on 1/1/2021 in Belgium in a phased manner [12].

#### 2.5.2. Severity Scores

In addition to other clinical outcomes of interest, such as intensive care unit (ICU) admission or length of stay (LOS), we included ‘COVID-19 severity’ as one of the variables. As the severity of COVID-19 can vary with time during hospitalisation, we decided to take the worst score for each patient’s stay. We chose to use the National Institute of Health (NIH) severity scores, as shown in Figure 1. All patients were therefore assigned a single NIH severity score [13,14].

### 2.6. Statistical Analyses

Descriptive statistics, numbers (proportions), and means (standard deviation) were used to summarise the cohort’s characteristics per variant wave. We then performed Pearson’s chi-square (χ^2^) tests for categorical variables and Kruskal Wallis tests for continuous variables to establish whether there was any difference between any of the four periods of interest (pre-VOC, Alpha, Delta, and Omicron periods).

For a selection of variables for which a difference was observed between variant waves on χ^2^ analyses, we performed χ^2^ or Dunn’s post-hoc tests to determine where the differences were present. This selection was based on evidence from the literature and clinical expertise. All tests were performed using SPSS (IBM Corp., released 2021). IBM SPSS Statistics for Windows, Version 28.0. Armonk, NY, USA: IBM Corp.).

## 3. Results

In total, we included 1326 patients.

### 3.1. Demographic Data and Co-Morbidities, per Variant Wave

The 1326 patients’ demographics and co-morbidities are shown in Table 1.

We discovered that the patients admitted to our hospital for COVID-19 during the Delta-wave were younger than those admitted during the pre-VOC and the Omicron periods (*p* < 0.001). Patients in the Alpha wave were also younger than those in the Omicron wave (*p* = 0.008). Patients with lung pathology made up a smaller proportion of COVID-19 hospitalisations in the earliest pre-VOC period than in the later Delta and Omicron waves. Patients with active cancer made up a larger proportion of COVID-19 admissions in the Omicron period compared to pre-VOC and Delta. Patients who were admitted during the Omicron wave had a higher score for frailty than those who were admitted during any of the other waves.

### 3.2. Vaccination Status and Treatment during Admission, per Variant Wave

The 1326 patients’ vaccination status and treatments received during admission are shown in Table 2.

The number of unvaccinated patients varied by period. The proportion of partially vaccinated people was highest in the Alpha wave compared to all other waves For fully vaccinated individuals, a difference was seen between pre-VOC vs. Alpha vs. Delta + Omicron periods. The proportion of patients who had a booster was highest for Omicron, then for Delta, compared to the pre-VOC period; the proportion in the Omicron period was also higher than in the Alpha period. The vaccination data are summarised in Figure 2a.

Patients were placed in the prone position more often during the Alpha and Delta periods compared to pre-VOC. There was no difference between the Omicron and the other periods.

### 3.3. Patients’ Outcomes and Severity Scores per Variant Wave

The 1326 patients’ outcomes and severity scores are shown in Table 3.

While there was no difference in the proportion of COVID-19 patients dying in hospitals across the waves, COVID-19 was the sole cause of death for a smaller proportion of patients during the Delta wave than during previous waves.

Additionally, the length of stay was different during the Alpha period and the Delta period. During the Delta period, it was shorter. When compared to the pre-VOC and Omicron periods, patients hospitalised during the Alpha wave had a longer ICU stay.

Regarding NIH severity scores, differences exist among the ‘severe’ and ‘critical’ groups. The highest proportion of ‘critical’ patients (47.7%) was observed during the Alpha period, while the Omicron wave saw the highest proportion of patients in the ‘severe’ category (61.6%). The NIH severity scores per wave are shown in Figure 2b.

## 4. Discussion

We were able to study a large group of COVID-19 patients (1326) during four distinct variant waves spanning 2 years of the SARS-CoV-2 pandemic against a background of adaptive personal protection, treatment guidelines, and vaccination rollout in Belgium. The characteristics of hospitalised patients with COVID-19 changed profoundly from one epidemic wave to the next.

In particular, **clinical characteristics** varied very much per wave. Patients admitted to our hospital for COVID-19 were the youngest during the Delta wave. This may be explained by the fact that many older adults had already been infected or vaccinated by then. In contrast, the later Omicron wave again affected older adults, as described in other publications [15]. This is probably due to opposite factors: vaccines and natural immunity were both declining at the time. Even if most age groups were well-vaccinated when the Omicron wave hit Belgium [12], the more fragile adults, mostly the elderly, became the most susceptible group again, even to a less pathogenic virus.

Similarly, the proportion of inpatients with lung disease or cancer was higher in the later waves compared to the pre-VOC period. This could be because the large first wave of infections occurred in a totally COVID-19-‘naïve’ population, presenting a wide range of risk factors for severe COVID-19, among which lung disease or cancer, but each in a much smaller proportion than in the later waves. Immune escape also came into play, making those more fragile patients more susceptible, despite a “less virulent” virus. The higher clinical frailty scores in the latest Omicron wave are another clue that COVID-19 inpatients changed over time. These results contrast with earlier papers from South Africa at the start of the Omicron wave [16]. Such epidemiological differences should warn us that the same virus can lead to different findings in different settings or stages of the pandemic.

Looking at **vaccination status**, first focusing on the Omicron wave, unvaccinated patients (35.6%) made up a large proportion of admitted patients during the latest Omicron period. This is a much larger group than the proportion of unvaccinated adults in the general Belgian population at that time (10.3%) [10]. This could be in part due to regional variations in vaccination uptake in Belgium so that 24.11% of adults in the Brussels region were unvaccinated on 5/2/2022 [12]. An even larger proportion of patients admitted in the Omicron period were fully vaccinated or ‘boosted’, compared to previous waves—and nevertheless needed hospital admission. Waning immunity undoubtedly played a role here, particularly in the group that had not yet received the booster [17], as their primary vaccination course had occurred earlier. Furthermore, vaccines used up until then are now known not to have covered the Omicron variant as well. In addition, most patients in that period had more co-morbidities; they are more likely to have been immunosuppressed and therefore less likely to have responded to the vaccines in the first place.

Despite its reduced intrinsic severity [18] and in a context of decreased sense of urgency to get vaccinated or to use preventive measures amongst the population, the Omicron variant was therefore the cause of a significant number of hospitalisations.

Regarding **severity outcome measures**, variables reflecting more severe clinical states were more frequent during the Alpha period: prone positioning (also more frequent during the Delta wave), LOS, ICU admission, and LOS on ICU. The NIH severity score [13] was a valuable tool in this context and showed that patients with the highest NIH severity score, i.e., ‘critical’, also made up the largest group of patients admitted during the Alpha wave (Figure 1b).

It may seem paradoxical that pre-VOC COVID-19 cases generally showed less severe clinical outcomes than those presented in the Alpha and Delta waves. Unlike those exposed to subsequent waves, no pre-VOC cases were vaccinated or had previous SARS-CoV-2 infections that would be expected to protect against more severe disease; optimal therapeutic protocols were not available; and these patients tended to be older. Different factors may explain this. First, vaccination rates in the general adult population at the time of the Alpha wave were still quite low, and combined with lockdowns imposed in Belgium that did not allow the virus to circulate to increase general protection; the pool of susceptible patients was still very large indeed. Second, the Alpha wave was the first breakthrough variant of the original coronavirus, and available data suggest that the Alpha (B.1.1.7) and Delta (B.1.617.2) variants of SARS-CoV-2 were associated with higher transmissibility and disease severity [19,20,21] compared to the pre-existing variants. Third, Optiflow, a high-flow oxygenation technique, was used more widely after the pre-VOC period and may have skewed the results towards longer ICU and total LOS as well as towards the ‘critical’ severity category in the later waves [20,22]. Fourth, another reason might be more subjective. At the start of the pandemic, when the clinical course of COVID-19 was still completely unknown to clinicians, an excess of admissions may have taken place through fear of further deterioration and despite admission criteria being constant. Finally, we do not believe hydroxychloroquine (HC) played a role in the apparent less severe pre-VOC period, even if HC users in our cohort seemed to have lower severity scores and to die less (data not shown). This is not enough hard evidence to draw any conclusions due to the observational nature of our study. We stopped administering HC as worldwide evidence confirmed that it was not effective against COVID-19 [23,24].

The proportion of ‘critical’ patients, using NIH criteria, decreased with time from the Alpha to the Omicron wave, as described elsewhere [25]. In contrast, the proportion of patients with ‘severe’ NIH scores was the highest during the Omicron wave (Figure 1), compared to other waves. These are still quite sick patients, and again, it may seem counterintuitive given the ‘globally less virulent’ Omicron variant [26]. On the one hand, a large portion of the population was unvaccinated, waning immunity was present, and immunosuppressed patients [27] responded less well to vaccines, which were also less effective against the Omicron variant. On the other hand, vaccinated individuals were being protected from getting even sicker (thus not progressing to the ‘critical’ category). Surprisingly, the overall death rate across the four waves was not different, highlighting the level of illness of our cohort, even during the Omicron wave. Another study found no difference in in-hospital mortality between the Delta and Omicron waves [28].

One of the great **strengths** of our study is its data source: our *UZB SARI-Registry* is an automated registry that extracts detailed clinical data from patients’ Electronic Health Records (EHR). It was set up across the whole hospital starting from the onset of the pandemic in March 2020 and is therefore very comprehensive and complete, down to clinical details and vaccination status. The extraction process is also very safe, requiring only a few manual steps. Such a detailed clinical registry remains the exception [16]. It allowed us to study the whole of our hospitalised COVID-19 population across four variant waves.

To address some of the limitations of our study, we chose well-**defined periods** of viral circulation of specific variants of the SARS-CoV-2 virus in Belgium. Although sequencing capacity increased with time in our centre, we chose not to use individual variant identification, as this would have led to excessive missing data. We are confident that the Belgian national surveillance data reflects our catchment area’s epidemiology, as shown in another project published by our group recently [29]. We were also reassured to read a recent paper by Meurisse et al. [11], who found that exposure ascertainment based on time periods leads to the most precise results when studying severity per variant wave. We, therefore, chose periods in which each variant was circulating at levels of at least 80%, based on national COVID-19 surveillance data. It meant we did not include the Beta VOC period because its dominance never reached 80% in Belgium.

The paper by Meurisse et al. [11] also discusses the importance of **defining disease severity** outcomes in advance when studying severity in COVID-19 patients. First, our study focused on inpatients. Second, we decided to include only patients for whom COVID-19 was the main reason for admission. Some researchers have used similar basic inclusion criteria [16] but could not always tell whether COVID-19 was the main reason for admission [9], making our data more precise. Third, we used classic severity outcomes once in the hospital (ICU admission, etc.), but we also looked at the NIH severity score to reflect each patient’s worse state. Indeed, not all patients with the most severe disease are admitted to the ICU, for instance, based on patients’ and clinicians’ choices, thus introducing some bias, which can be evened out using a score [13].

Other known challenges inherent to observational studies are **confounding factors fluctuating over time**. First of all, the state of a health system during the COVID-19 pandemic can have an impact on whom to admit and, in turn, on patients’ outcomes [11]. The Belgian system was never completely overwhelmed during the pandemic, and admission criteria remained constant. Second, medical management of COVID-19 and severe COVID-19 evolved with time. Dexamethasone became available during the pre-VOC era, followed by antivirals and immunomodulating agents, leading to evolving treatment guidelines. This is why we did not focus on variables for which changes across waves could be explained by changes in medical practice. Finally, vaccines became available at the start of 2021 in Belgium. This introduced a major shift in susceptibility to SARS-CoV-2, which decreased as vaccination was rolled out [12]. Immunity from vaccination and natural infection increased and then waned again. Similar variations took place after every booster thereafter (not included in our study). Our study was not designed to control for all of these factors, but it was crucial to keep them in mind in the interpretation of our results.

We tried to keep missing data to a minimum by manually checking and completing those when feasible. The number of missing data records can be calculated in the results tables. Our study was mono-centric, so the data may not be as representative of larger populations as in multi-centric studies. On the other hand, changes in medical and infection control practices were implemented uniformly and hospital-wide. A multi-centre study would have had to deal with those extra challenges.

## 5. Conclusions

Our study shows that patients admitted to our hospital for COVID-19 during four different epidemic waves in Belgium differed in clinical and outcome profiles, with patients in the Alpha wave being the most severely ill. Host factors, vaccination, and the type of infecting SARS-CoV-2 variant most probably accounted for those differences.

The great variation in patients’ profiles across the epidemic waves is one of the lessons learned for future pandemics so that public health and medical decision-makers must remain vigilant and ready to adapt to fast-changing realities. In order to inform policymakers, high-quality data must be collected in a safe manner and in real-time. Detailed clinical registries at the hospital level are an invaluable tool to provide such in-depth insights into pandemics.

## Figures and Tables

**Figure 1 viruses-15-00618-f001:**
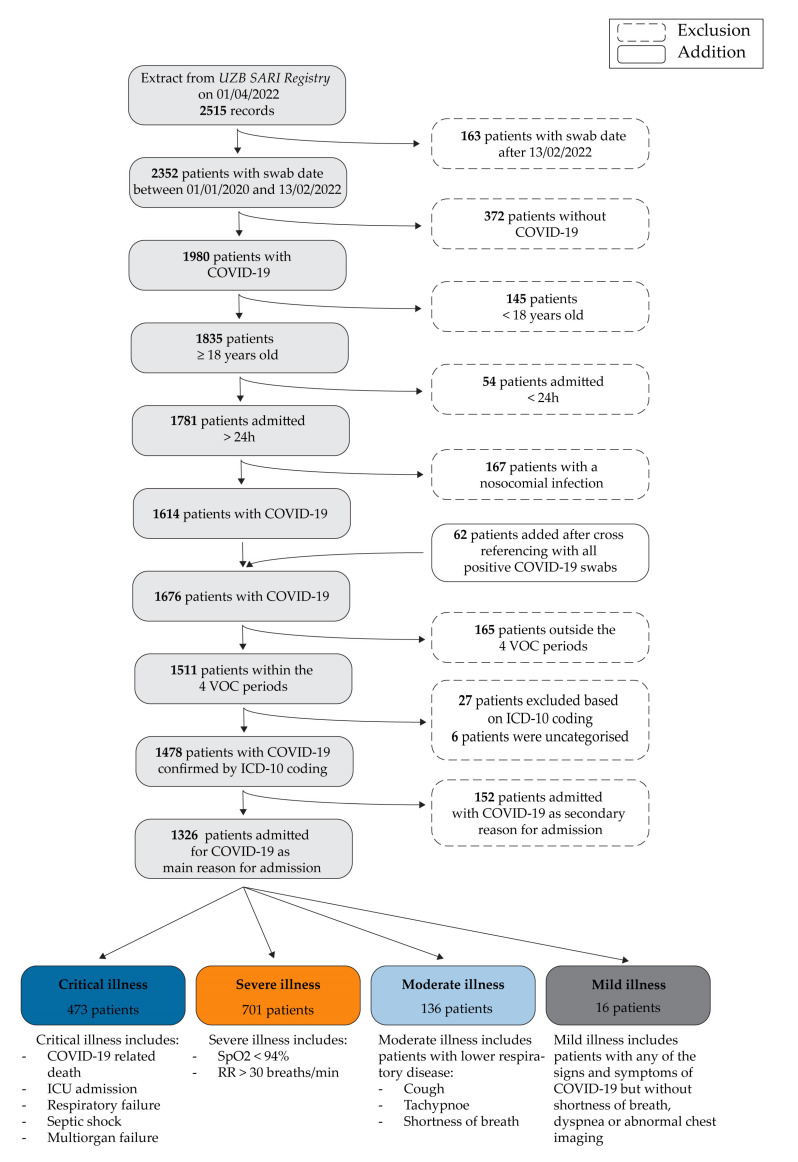
Patient selection steps (above) and NIH severity scores (below).

**Figure 2 viruses-15-00618-f002:**
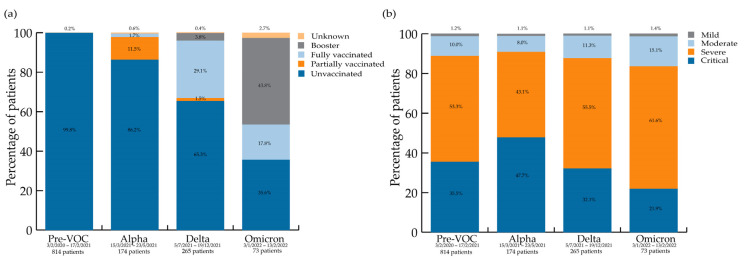
(**a**) COVID-19 vaccination status per variant wave; (**b**) NIH severity scores per variant wave.

**Table 1 viruses-15-00618-t001:** Demographic data and co-morbidities per variant wave.

Period of Admission	Pre-VOC	Alpha	Delta	Omicron	*p*-Value
**Number of patients** (% within wave)	**814**	**174**	**265**	**73**	**-**
**Women**	341 (41.9%)	72 (41.4%)	121 (45.7%)	36 (49.3%)	0.467
**Ethnicity §**					<0.001
**Age**	65.3 ± 17.3	62.1 ± 16.4	58.6 ± 18.1	69.6 ± 19.1	<0.001 *
**Smoking status:**					0.010
Never	289 (61.1%)	112 (73.7%)	150 (71.8%)	32 (65.1%)	
Former	149 (31.5%)	32 (21.1%)	46 (22.0%)	23 (37.7%)	
Current	35 (7.4%)	11 (5.3%)	13 (6.2%)	6 (9.8%)	
**BMI §**	28.2 ± 5.7	28.9 ± 5.5	28.5 ± 6.1	27.7 ± 6.2	0.084
**Lung disease**	143 (17.7%)	38 (22.2%)	58 (26.2%)	26 (36.1%)	<0.001 *
**Cancer**	44 (5.5%)	15 (8.7%)	15 (6.8%)	14 (19.4%)	<0.001 *
**Hypertension**	352 (43.7%)	86 (50.0%)	92 (45.3%)	39 (54.2%)	0.198
**Dementia**	50 (6.2%)	4 (2.4%)	11 (6.3%)	3 (4.4%)	0.239
**Diabetes**	237 (29.3%)	49 (28.5%)	48 (23.4%)	18 (26.5%)	0.407
**Heart disease**	199 (24.7%)	42 (24.4%)	39 (19.7%)	24 (33.8%)	0.121
**Neuromuscular disorder**	26 (3.2%)	6 (3.6%)	5 (2.8%)	4 (5.8%)	0.675
**Renal disease**	125 (15.5%)	31 (18.0%)	26 (11.7%)	13 (18.3%)	0.286
**Rheumatological disease**	46 (6.1%)	5 (2.9%)	15 (6.8%)	4 (5.8%)	0.380
**Stroke**	28 (3.7%)	4 (2.4%)	6 (3.4%)	4 (6.1%)	0.590
**Clinical Frailty Scale**	3.6 ± 1.6	3.3 ± 1.7	4.6 ± 3.4	5.3 ± 2.9	<0.001 *

* = variables for which post-hoc analysis was performed; § = additional data to be found in Appendix A. Age, BMI, and Clinical Frailty Scale are expressed in mean ± SD (standard deviation).

**Table 2 viruses-15-00618-t002:** Vaccination status and in-hospital treatment, per variant wave.

Period of Admission	Pre-VOC	Alpha	Delta	Omicron	*p*-Value
**Number of patients** (% within wave)	814	174	265	73	-
**COVID-19 vaccination status**					<0.001 *
Not vaccinated	812 (99.8%)	150 (86.2%)	173 (65.3%)	26 (35.6%)	*
Partially vaccinated	2 (0.2%)	20 (11.5%)	4 (1.5%)	0 (0.0%)	*
Fully vaccinated	(0.0%)	3 (1.7%)	77 (29.1%)	13 (17.8%)	*
Booster	(0.0%)	(0.0%)	10 (3.8%)	32 (43.8%)	*
Unknown vaccination status	0 (0.0%)	1 (0.6%)	1 (0.4%)	2 (2.7%)	
**Specific COVID-19 treatments**					
**Nasal oxygen**	738 (90.7%)	159 (91.4%)	249 (94.0%)	70 (95.9%)	0.200
**Prone position**	49 (6.0%)	36 (20.7%)	33 (12.5%)	7 (9.6%)	<0.001 *
**Antiviral**	36 (4.7%)	4 (2.3%)	0 (0.0%)	0 (0.0%)	<0.001
**Hydroxychloroquine**	303 (39.5%)	0 (0.0%)	1 (0.4%)	1 (1.4%)	<0.001
**Corticosteroid**	326 (42.5%)	152 (87.9%)	233 (88.3%)	54 (76.1%)	<0.001
**Il6-blockade**	0 (0.0%)	1 (0.6%)	14 (5.3%)	6 (8.5%)	<0.001
**Antibiotics**	345 (45.0%)	63 (36.4%)	99 (37.5%)	38 (53.5%)	0.013
**LMWH**	331 (43.2%)	156 (90.2%)	231 (87.5%)	58 (81.7%)	<0.001

* = variables for which post-hoc analysis was performed; IL6 = interleukin-6 blockade; LMWH = low molecular weight heparin.

**Table 3 viruses-15-00618-t003:** Outcomes and severity scores, per variant wave.

Period of Admission	Pre-VOC	Alpha	Delta	Omicron	*p*-Value
**Number of patients** (% within wave)	814	174	265	73	-
**Outcome**					
Died	146 (17.9%)	27 (15.5%)	30 (11.3%)	10 (13.7%)	0.076
Discharged	668 (82.1%)	147 (84.5%)	235 (88.7%)	63 (86.3%)	
**Cause of death**					
COVID-19	143 (97.9%)	25 (96.2%)	14 (66.7%)	4 (100.0%)	<0.001
COVID-19 + other	1 (0.7%)	0 (0.0%)	6 (28.6%)	0 (0.0%)	
Other	2 (1.4%)	1 (3.8%)	1 (4.8%)	0 (0.0%)	
**LOS (in days)**	12.8 ± 21.7	18.7 ± 29.5	11.6 ± 14.7	12.9 ± 15.2	0.028 *
**ICU admission**	192 (23.6%)	62 (35.6%)	74 (27.9%)	10 (13.7%)	<0.001 *
**ICU LOS (in days)**	14.8 ± 20.1	22.4 ± 25.6	16.9 ± 19.2	19.3 ± 15.2	0.007 *
**NIH Severity score**					0.022 *
Mild	10 (1.2%)	2 (1.1%)	3 (1.1%)	1 (1.4%)	
Moderate	81 (10.0%)	14 (8.0%)	30 (11.3%)	11 (15.1%)	
Severe	434 (53.3%)	75 (43.1%)	147 (55.5%)	45 (61.6%)	
Critical	289 (35.5%)	83 (47.7%)	85 (32.1%)	16 (21.9%)	
**Complications**					
None	378 (49.3%)	64 (37.0%)	159 (60.2%)	32 (45.1%)	<0.001
ARDS	41 (5.0%)	22 (12.6%)	12 (4.5%)	1 (1.4%)	<0.001
Pneumonia (sec. bacterial)	82 (10.1%)	31 (17.8%)	24 (9.1%)	8 (11.0%)	0.018
Sepsis	35 (4.3%)	11 (6.3%)	3 (1.1%)	0 (0.0%)	0.008
Heart failure	41 (5.3%)	24 (13.9%)	6 (2.3%)	2 (2.8%)	<0.001
Multiorgan failure	7 (0.9%)	8 (4.6%)	1 (0.4%)	0 (0.0%)	<0.001
Acute renal injury	119 (14.6%)	17 (9.8%)	6 (2.3%)	1 (1.4%)	<0.001
Thrombotic event (DVT or PE)	21 (2.6%)	7 (4.0%)	6 (2.3%)	3 (4.1%)	0.605
Respiratory failure	97 (12.6%)	45 (26.0%)	28 (10.6%)	8 (11.3%)	<0.001

* = variables for which post-hoc analysis was performed; ‘COVID-19 + other’ = cause of death was COVID-19 and other causes, ‘Other’ = cause of death was not COVID-19; LOS = length of stay in hospital; ICU = intensive care unit; ARDS = acute respiratory distress syndrome; sec = secondary; DVT = deep venous thrombosis; PE = pulmonary embolism; LOS and ICU LOS are expressed in mean ± SD (standard deviation).

## Data Availability

We provide additional data in the Appendix A. For access to our full dataset, please contact us at the following mail address: sariregistry@uzbrussel.be.

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
