# Peer review of "Surfing the Waves: Differences in Hospitalised COVID-19 Patients across 4 Variant Waves in a Belgian University Hospital"

_viruses, 2023, doi:10.3390/v15030618_

Round 1

Author Response

Dear Reviewer 1, 

Thank you very much for your very constructive feedback on our paper. 

Please see the attachment for a point-by-point response to your comments. 

Yours faithfully, 

Dr Lucie Seyler

Reviewer 2 Report

Dear Authors

I would like to thank you for the opportunity of reviewing this interesting paper that is focused on a very remarkable and challenging topic that is a lively argument also in the daily clinical practice. The ongoing COVID-19 pandemic has overwhelmed healthcare systems globally, imposing serious health, economic and social effects on the population. 

Several previous published studies have focused on data spanning one or a couple of COVID-19 waves at a time, but data of the evolution of COVID-19 across more waves are scarce. The present study aims to analyze the clinical characteristics and severity of patients hospitalised for COVID-19 across the COVID-19 pandemic in a Belgian university hospital, investigating any potential shifts in patients’ profiles across the 4 most significant epidemic waves in the country.

This paper is pleasurable to read, although it suffers from some limitations that Authors can easily adjust in order to slightly improve their review making it more eligible for this important Journal. Furthermore, Authors can improve some section of the paper, adding information and including other important references about this topic that, in my opinion, should be cited and discussed. 

First of all, although language used is appropriate, I (I am not a native English speaker) recommend to Authors to obtain a certified native speaker with proficiencies in the scientific-medical field to complete properly this paper (if not jet done). Moreover, I recommend making a further revision of the manuscript to fix some small typing/language errors.

The title is clear and direct. Personally, I believe it could be improved and be more focused on results. For example: Surfing the waves: differences between hospitalised COVID-19 patients across variant waves in a Belgian university hospital”. 

The abstract is well structured and properly reflects the main text highlighting the most important aspects of this paper. However, I recommend to add which score system has been used also in the sentence “We compared clinical data and severity scores from all patients admitted for COVID-19 during 4 SARS-CoV-2 variant waves.” using the unabbreviated name (National Institute of Health (NIH) severity scores).

Authors did not correctly reported keywords from MeSH Browser. In particular, I checked for example “hospitalised patients” on MeSH Browser and this is not a KW. This is important, in my personal opinion, in order to increase the traceability of this paper (and consequently the possibility of the Journal to be cited by Readers and Stakeholders). I suggest the check of all KW. 

Although the introduction fits the context of the study, it is concise. Sometime, many concepts clearly explicated in an exhaustive introduction could help readers to become passionate about reading the paper and using it as a reference. In my opinion, it is important to underline that despite viral pneumonia has been recognized as the main clinical presentation of this disease and represents the main cause of its severity and mortality, COVID-19 infection can cause several complications also in other organs, with coagulation disorders (pulmonary embolism, venous thromboembolism, hemorrhages and acute ischemic stroke) and abdominal involvement (acute mesenteric ischemia, pancreatitis and acute kidney injury), especially in severely ill patients and those admitted to ICU [Radiol Med. 2022;127(4):369-382. doi:10.1007/s11547-022-01473-w] [Diagnostics (Basel). 2022;12(4):846. doi:10.3390/diagnostics12040846]. Please, cite these articles and introduce these important aspects in this section.

Moreover, regarding the sentence “Many published studies have focused on data spanning one or a couple of COVID-19 waves at a time [3]”, there are studies that have investigated also the differences in chest CT imaging appearance between waves [Emerg Radiol. 2021 Dec;28(6):1055-1061. doi: 10.1007/s10140-021-01937-y.] [Cureus. 2022;14(1):e21656. doi:10.7759/cureus.21656]. Please also cite these papers.

In Materials & Methods, regarding the section 2.4. Patient selection, I believe Figure 1A could be included in the main text. Moreover, I suggest to explicit whether all patients included in the study underwent a swab test with PCR analysis.

I suggest transforming the NIH severity score for COVID-19 section into a figure.

In table 2, please include all the different treatments under a Title (such as “Specific Treatment”).

In my opinion, the topics of the discussion are very beautiful but overall, the discussion is too long. The authors must decide to reduce the text which otherwise becomes difficult to read. 

In the limits section, please state that beta VOC wave did not reach 80% dominance at any point and was therefore not included in the study.

Finally, I think references should be reformat as suggested by Viruses Author’s guidelines (Author 1, A.B.; Author 2, C.D. Title of the article. Abbreviated Journal Name YearVolume, page range)

Author Response

Dear Reviewer 2, 

We are very grateful for all your comments and suggestions, and we have adapted our manuscript accordingly. 

Thank you again, 

Yours faithfully, 

Dr Lucie Seyler

Round 2

Reviewer 2 Report

Authors addressed raised points appropriately.